# Anatomy education potential of the first digital twin of a Korean cadaver

**Seo Yi Choi**[1]☉, **Dong Hyeok Choi**[2,3,4]☉, **Sung Ho Cho**[5], **So Hyun Ahn**[1,6,7]*☉, **Seung Ho Han**[8,9]*☉

1 Department of Biomedical Engineering, College of Medicine, Ewha Womans University Seoul, Seoul, Republic of Korea, 2 Medical Physics and Biomedical Engineering Lab (MPBEL), College of Medicine, Yonsei University, Seoul, Republic of Korea, 3 Department of Radiation Oncology, College of Medicine, Yonsei Cancer Center, Heavy Ion Therapy Research Institute, Yonsei University Seoul, Seoul, Republic of Korea, 4 Department of Medicine, College of Medicine, Yonsei University Seoul, Seoul, Republic of Korea, 5 REMEDI Research and Development Center, Seoul, Republic of Korea, 6 Ewha Medical Artificial Intelligence Research Institute, College of Medicine, Ewha Womans University, Seoul, Republic of Korea, 7 Ewha Medical Research Institute, College of Medicine, Ewha Womans University, Seoul, Republic of Korea, 8 Human Bio Information Group, Ewha Womans University Seoul Hospital, Seoul, Republic of Korea, 9 Department of Anatomy, College of Medicine, Ewha Womans University, Seoul, Republic of Korea

☉ These authors contributed equally to this study.
* mpsohyun@ewha.ac.kr (SHA); sanford@ewha.ac.kr (SHH)

## Abstract

The objective of this study is to explore innovative integration within the field of anatomy education by leveraging HoloLens 2 Augmented Reality Head-Mounted Display (AR HMD) technology and real-time cloud rendering. Initial 3D datasets, comprising extensive anatomical information for each bone, were obtained through the 3D scanning of a full-body cadaver of Korean male origin. Subsequently, these datasets underwent refinement processes aimed at enhancing visual fidelity and optimizing polygon counts, utilizing Blender software. Unity was employed for the development of the Metaverse platform, incorporating tailored 3D User Experience (UX) and User Interface (UI) components to facilitate interactive anatomy education via imported cadaver models. Integration with real-time remote rendering cloud servers, such as Azure, was implemented to augment the performance and rendering capabilities of the HoloLens 2 AR HMD. The extended reality (XR) content uses the Photon Cloud network for real-time data synchronization and HoloLens 2 voice functionality. The metaverse platform supports user interaction through room creation and joining, with various tools for bone manipulation, color differentiation, and surface output. Collaboration features enable sharing and synchronization of model states. The study highlights the importance of technological innovation in anatomy education for future medical professionals. The proposed content aims to address limitations of traditional methods and enhance learning experiences. Continued efforts in developing and improving such technologies are crucial to equip learners with essential skills for adaptation in the evolving healthcare landscape.

**Data availability statement:** "All relevant data are within the paper and its Supporting Information files."

**Funding:** This research was supported by a grant of the Korea Health Technology R&D Project through the Korea Health Industry Development Institute (KHIDI), funded by the Ministry of Health & Welfare, Republic of Korea (grant number: RS-2023-KH134708).

**Competing interests:** The authors declare that they have no conflicts of interest.

## Introduction

Anatomy education plays a crucial role in medical training by providing a foundational understanding of the human anatomy [1]. Anatomy education plays a fundamental component not only for aspiring surgical doctors but also for students in nursing departments and other healthcare fields [2,3]. However, traditional methods of teaching anatomy, such as cadaver dissection, have raised ethical concerns, financial constraints, and other limitations. Cadavers are not always accessible as there are challenges in specimen availability, preservation, maintenance, as well as the high cost of implementation [4–6]. The declining trend in cadaver use raises concerns about the reduced opportunities for aspiring medical students to gain practical experience in dissection and gross anatomy [7, 8]. Social distancing and restrictions as responses to pandemic situations, like the recent COVID-19 pandemic, has further exacerbated this issue, leading to a significant reduction in the use of cadavers in anatomy education. A survey conducted among anatomy educators in the United States has shown a decline in using cadavers, specifically 39% in gross anatomy courses and 43% in other anatomy courses during the pandemic [9].

In time, advancements in technology have provided opportunities to develop integrative approaches that address ethical concerns and have helped to improve the level of comprehensiveness in anatomy education for students [10,11]. The use of imaging technologies such as MRI and CT scans has enabled researchers to visualize the internal structures of the human body with unprecedented detail. Additionally, virtual reality (VR) and augmented reality (AR) has been used to create interactive 3D models that allow students to explore the body and its systems in an engaging and immersive way. The integration of these technologies has revolutionized the delivery of medical education and expanded the possibilities for remote learning and digital resources [12–18].

The effects of COVID-19 have even accelerated the need for integrating the latest interactive technology in medical education, highlighting the importance of innovative solutions such as extended reality (XR) technology and the metaverse. The use of XR and the metaverse in clinical practice education is gaining popularity in addressing these challenges [19]. For example, the University of Northampton is using VR technology to provide nursing education [20], while the University of Newcastle is using VR simulations for childbirth education [21]. As such, the integration of XR technology in medical education offers numerous advantages, such as enhancing student learning outcomes, providing hands-on experience within a safe simulated environment, promoting interprofessional education, saving costs and time, and the potential to revolutionize the way medical education is delivered [22].

Although the integration of XR technology and attempts in anatomy education offer numerous advantages, several challenges need to be addressed [23]. Among these challenges, the integrity of the anatomy content is a primary concern due to various factors such as the quality of the original scan and the modeling design [24]. Inaccuracies in the 3D models may result in misinterpretations of anatomical structures, which can be particularly problematic in medical education where precision is paramount. Another challenge is the amount of data required in creating a detailed and accurate representation of anatomical structures. The creation of high-resolution 3D models for XR anatomy education requires a fast-paced render system to handle heavy data while maintaining smooth performance for immersive interactivity and an optimal user experience. Consequently, the development of such content necessitates the adoption of specialized solutions and techniques to ensure seamless operation.

While the Metaverse/XR technology offers an immersive, interactive experience that allows for detailed exploration and manipulation of anatomical structures, it does not provide the physical feedback and tactile experience that a 3D printed model can offer. The physical 3D printed models allow students and professionals to physically handle and inspect anatomical

structures, which can enhance their understanding of spatial relationships and material properties. This is particularly valuable in medical education, where hands-on practice is crucial. Moreover, the 3D DT provides an ethical and readily available alternative to traditional cadaver use, which is often limited by availability and ethical concerns. The physical models can be produced in unlimited quantities, providing consistent, high-quality educational resources that can be accessed without the logistical and ethical issues associated with cadaveric specimens. Thus, the use of both 3D printed models and the Metaverse/XR platform offers a comprehensive educational experience, leveraging the strengths of both physical and virtual modalities to enhance learning outcomes. To provide a comprehensive educational experience that improves learning outcomes by taking advantage of physical and virtual modalities, we 3D printed human bones.

This research endeavors to propose optimal solutions and unique methodologies to the challenges encountered in XR anatomy education by developing a 3D Digital Twin (DT) of a Korean Standard Full-body Male Cadaver (KSMC) in metaverse. Key aspects include the use HoloLens 2 Augmented Reality Head-Mounted Display (AR HMD) technology and real-time cloud rendering for an immersive and interactive educational experience. Initial 3D datasets, comprising extensive anatomical information for each bone, were obtained through the 3D scanning of a full-body cadaver of a Korean male origin.

This initial study focuses on the digital twin created from the bone scan of the Korean cadaver, with complementary studies planned to include flesh, organs, and other body parts.

The process of creating a 3D DT of a KSMC involved acquiring raw 3D object files for each bone using a 3D surface scanner. Subsequently, each 3D object file underwent refinement for visual accuracy while maintaining the integrity of the anatomical features of the original cadaver. This was achieved by adjusting the number of polygons to the optimal level using *Blender*, a 3D modeling software. The resulting model was then integrated into a metaverse platform built using Unity, which consisted of UX/UI elements designed to facilitate the immersive user experience. Furthermore, *Azure*, a cloud computing platform, was employed to provide real-time remote rendering services, which addressed the challenge of rendering high-resolution 3D models and alleviated the performance burden on the *HoloLens 2 AR HMD*. This component facilitated not only the individual user's immersive experience but also allowed multiple users to interact and communicate with the digital twin cadaver in real-time, presenting a unique and innovative approach to anatomy education. In essence, this research aims to pioneer XR anatomy education developing a 3D DT of a KSMC in the metaverse, introducing transformative and collaborative approaches to current challenges and future possibilities in medical education and telemedicine.

## Materials and methods

The process of creating a virtual 3D DT of a full-body KSMC involved collecting raw 3D object files for each bone using a 3D surface scanner. Subsequently, each 3D object file underwent refinement for visual accuracy while maintaining the integrity of the anatomical features of the original cadaver. This was achieved by adjusting the number of polygons to the optimal level using Blender and Maya, the two most widely used professional 3D modeling software. The resulting model was then integrated into a metaverse platform, built using Unity, which consisted of UI/UX elements designed to facilitate the immersive user experience. To enhance the interactivity and real-time communication capabilities of the metaverse platform, Microsoft Mixed Reality Toolkit (MRTK) and Photon Network were also used with Unity. It is often challenging for data-intensive content to perform seamlessly at high resolutions. However, this study utilized Microsoft's Azure cloud computing platform to provide real-time remote rendering services, enabling the rendering of high-resolution 3D models while alleviating the

performance burden on the HoloLens 2 AR HMD. Not only do these components enhance the individual user's immersive experience but also allow multiple users to interact and communicate with the DT in real-time, presenting a unique and innovative approach to anatomy education and telemedicine. Once the 3D DT of the KSMC was successfully imported into the developed Metaverse platform, the pilot test feedback was gathered from anonymous participants in the medical field, including company employees, students and professors from medical and nursing schools. They included open-ended survey from interests, evaluations and adoptability, and suggestions for improvements, which will be found throughout the article. No participants were required to answer all categories but required to answer at least one.

## From cadaver to a 3D digital twin cadaver

**Collecting and scanning.** To create a 3D DT of a KSMC, we followed a multi-step process that began with the ethical procurement of a full-body cadaver from a certified Korean institution. A total of 206 individual bones of a male cadaver were donated to the Department of Anatomy of Catholic University of Korea and were then shared with Ewha Woman's University for a full 3D scan.

For the 3D scanning process, we used a Leica BLK360 3D laser scanner, which offers several advantages over other 3D scanning methods. One of the main benefits of the BLK360 scanner is that it does not harm the cadaver, making it an ethical choice for capturing high-quality 3D scans. Additionally, the scanner is fast, accurate, and captures data with a high level of detail. To scan the cadaver body, we first placed it on a stable platform, and then used the BLK360 scanner to capture a series of 3D scans from multiple angles. These scans were then combined and processed using specialized software to create a complete and accurate 3D model of the cadaver.

The use of the Leica BLK360 scanner offers several advantages over other scanning methods. First, it provides high accuracy and resolution, which is essential for capturing the intricate details of the anatomy. Additionally, BLK360 scanner is a quick and efficient method, reducing the time required for the scanning process compared to other traditional methods. Most importantly, using this method allowed us to maintain the ethical standards of cadaver use, as the cadaver was kept intact and preserved for its intended purpose of donation.

**Refining the mesh.** After scanning the bones, each was then refined using 3D modeling software, called Blender and Maya. Refining a 3D scan in Blender or Maya is a crucial step as we develop anatomical content for the metaverse. This process involved using various tools and techniques to fill in any missing or incomplete parts of the model that may have been lost during the scanning process. To begin, the 3D scan is imported into Blender using the "Import" option in the "File" menu. Once imported, the model is carefully examined to identify any missing or incomplete parts using the "X-Ray" option in the "Viewport Overlays" menu. Using the Sculpt Mode in Blender, the model can be manually sculpted and shaped to fill in any missing details or smooth out rough areas.

**Retopologizing the mesh.** Retopology techniques can be used to create a new, cleaner mesh over the existing mesh of the scan. This can make it easier to work with and modify, as well as provide a more accurate representation of the scanned object. During the scanning process, it was discovered that some of the bones in the original cadaver exhibited synostosis, such as the middle and end phalanges of both fifth metatarsal bones, and the sacrum and coccyx. To address this issue, we consulted various medical resources and anatomical atlases as references and employed retopology techniques.

The fused bones were treated as single objects, and the six ear bones were combined with the 22 cranium bones for grouping purposes. After the mesh was retopologized, we utilized Mesh Modeling tools to further refine and shape the model. Specifically, these tools were

used to add or subtract polygons, create new shapes, or modify existing ones, especially for extremely thin layers or deep holes in the bones that impeded proper 3D modeling.

**Optimizing polygons.** In addition to refining the mesh of the 3D model, it is essential to optimize the number of polygons in the 3D models, especially for a real-time interactive experience. The number of polygons in a 3D model has a direct impact on its processing power and memory requirements for proper rendering. As the number of polygons increases, the performance of the virtual environment can significantly slow down. Therefore, it is crucial to optimize the number of polygons in 3D models to ensure smooth rendering and efficient performance in the metaverse.

We chose to optimize the number of polygons in the 3D objects to use techniques such as decimation, which involves reducing the number of polygons in a model while preserving its overall shape and structure. An important consideration when optimizing the number of polygons is to balance the level of detail required for specific use cases. Since this development creates anatomical content for educational purposes, it requires a higher level of detail and intricacy.

## The architecture to facilitate the 3D DT in metaverse

**HoloLens 2.** HoloLens is a mixed-reality device that allows users to interact with holographic objects and digital content in real-world environments. Table 1 shows HoloLens 2 specifications. This provides a more immersive and engaging experience than traditional 2D screens or virtual reality devices. HoloLens transforms anatomical content into an immersive, dynamic experience, as users can manipulate the organs, bones, and any anatomical structures of the 3D DT with simple hand gestures and voice commands, allowing them to explore and learn at their own pace. HoloLens can also provide real-time feedback and guidance, enhancing the learning experience and providing a more hands-on approach to education.

**Table 1. Specification of HoloLens 2.**

| Items | Function | Features |
|---|---|---|
| Sensors | Head Tracking | 4 visible light cameras |
| | Eye Tracking | 2 infrared cameras |
| | Camera | 8MP photos, 1080p 30 video |
| User recognition | Hand Tracking | Hand movements for direct manipulation |
| | Eye Tracking | Real-time eye tracking |
| Computing and connectivity | SoC (System on a Chip) | Qualcomm Snapdragon 850 computing platform |
| | HPU (Holographic Processing Unit) | 2nd generation custom holographic processing unit |
| | Memory | 4GB LPDDR4x system DRAM |
| | Storage | 64GB UFS 2.1 storage |
| | Wi-Fi | Wi-Fi 5 (802.11ac 2x2) |
| | Bluetooth | Bluetooth 5.0 |
| | USB | USB-C Type |
| Features | Wide Field of View | Doubled field of view compared to the previous HoloLens |
| | | High resolution allowing reading of font sizes down to 8pt |
| | Hand Tracking | Natural hologram tracking, including grabbing, touching, and moving in a controller-free HMD |
| | | Recognizes user's hands for projection on the screen, currently with the highest hand tracking accuracy |
| | Voice Support | Voice control available when hands are not in use |
| | Eye Tracking | Recognizes the user's gaze |
| | Spatial Mapping | Smoothly maps the physical environment, enabling fixing digital content anywhere on objects or surfaces |

**Developing a metaverse platform.** Once the 3D models were complete, scripting is required to create an interactive model. As we utilized Unity, which is best compatible with the HoloLens2, and C# as it is a commonly used scripting language in Unity. By writing scripts, the model can be manipulated by the user. For example, the user can rotate the model or select a specific body part to receive additional information about it. In addition, we incorporated Mixed Reality Toolkit (MRTK), which is a collection of scripts, components, and assets that simplifies the development of mixed reality applications, to include pre-built components such as buttons and sliders, as well as tools for input management and spatial mapping.

**Developing real-time interaction features between users.** For additional functionalities such as real-time interaction among multiple users, we integrated Photon Network, a real-time multiplayer network engine designed for Unity. This integration adds to the collaborative learning experience and facilitates teaching scenarios. These tools provide essential components for creating realistic 3D DT models and simulations, enabling seamless mixed reality experiences, real-time collaboration and conference capabilities. As such, 3D DT models can be created with both realistic precision and computational efficiency.

**Enhancing XR content with remote rendering.** Remote rendering can be a creative way to develop Unity programs for high-quality 3D anatomy content, as it enables the rendering of complex and data-intensive 3D models without the need for expensive local hardware or extensive processing power.

With remote rendering, the 3D model is sent to a remote server, or a cloud-based service. The rendered images are then transmitted back to the user's device, allowing smooth and high-quality visualization of the 3D anatomy content. This approach offers developers the advantage of ensuring optimal performance of their Unity programs across various devices, from low-powered laptops to high-end workstations. In addition, remote rendering can substantially cut the amount of the time and cost associated in developing and deploying Unity programs that feature data-intensive content like the interactive 3D DT models. By using cloud-based rendering services, developers can avoid the need for costly investments and focus on designing the most compelling interactive content for their users.

Overall, remote rendering serves as a powerful tool for developing Unity programs with smooth and high-quality 3D anatomy content. By leveraging the power of remote servers and cloud-based rendering services, developers can create immersive and engaging programs that provide a unique and valuable learning experience for students and professionals in the field of anatomy.

## Pilot test feedback

The Metaverse platform and the 3D DT content developed in our study underwent a thorough process of gathering pilot test feedback from anonymous participants in the medical field, including company employees, students and professors from medical and nursing schools. This feedback was collected through an open-ended survey covering general inquiries and interests, evaluations of quality and potential adoptability, and suggestions for further improvements. No participants were required to answer all categories, resulting in some respondents addressing one, two, or all of the topics.

## Results

### 3D digital twin cadaver

The 3D DT of a KSMC, with its meticulously scanned and refined data, was 3D printed using the internal filament removal method, to ensure both durability and accuracy. The model, with a size of 45 cm, as shown in Fig 1 (a), and the assembled model is depicted in Fig 1 (b),

suggests opening new opportunities for students and professionals to have hands-on tactile learning experiences with cadaver unlike traditional cadaver experiences, which are limited in availability and often faced with ethical issues. As such, it addresses that the 3D DT not only provides users with unlimited access to a cadaver model for education but also allows accessibility to unlimited physical copies of the meticulous 3D DT that students and professionals can physically manipulate and grow a deeper realistic understanding of.

## Architecture for the metaverse platform

The feasibility of data heavy XR content in real-time collaborative digital space, like the Metaverse, can only be justified through creative solutions like Photon Cloud Network, which synchronizes inter-user data. In this Metaverse platform, however, there are more to the architecture beyond Photon Cloud Network. An important feature for real-time sharing is identified, as 'REALTIME,' with the exact status of objects. The 'Add-on Voice' App ID key for

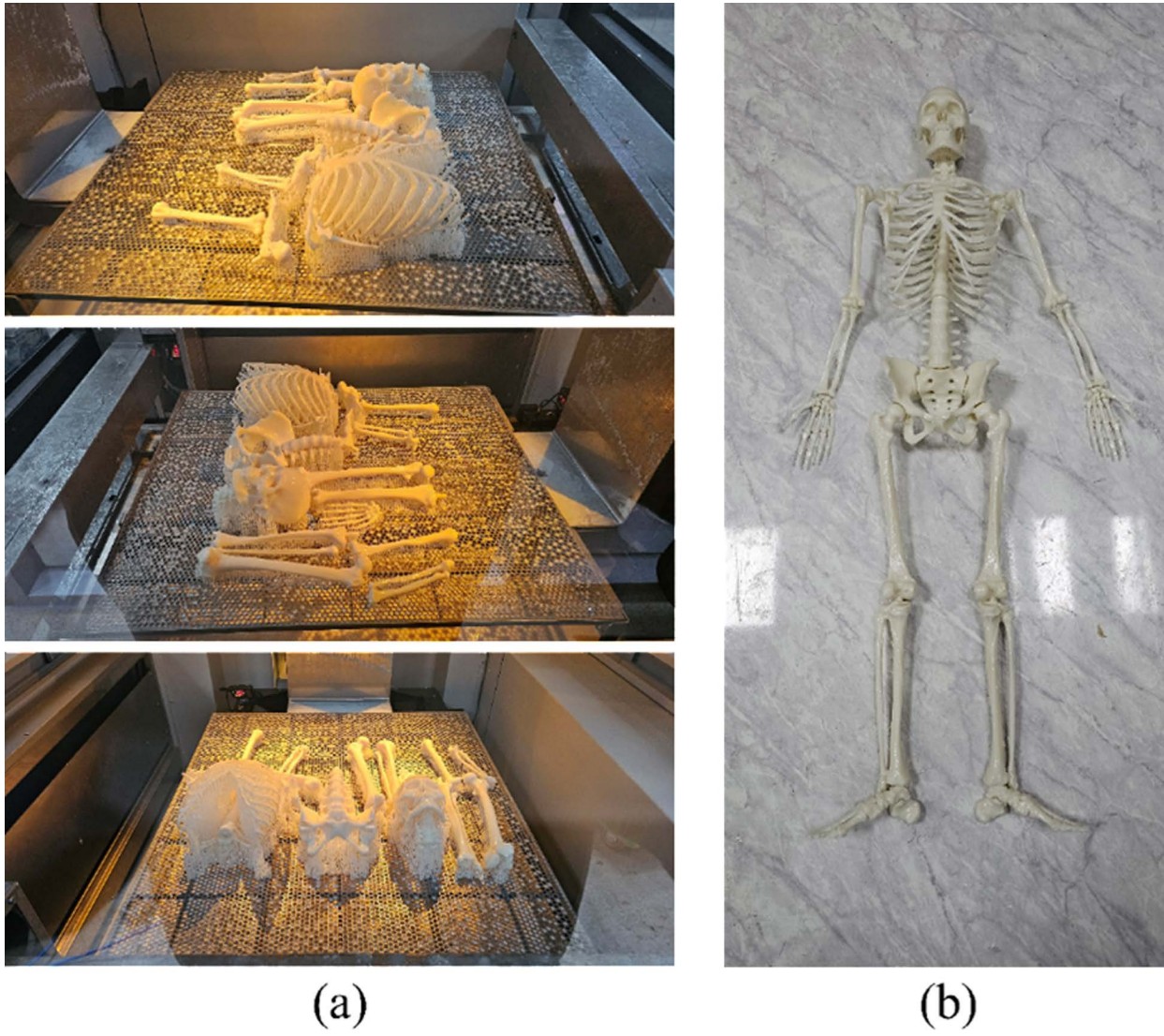

(a) (b)

**Fig 1. Bones printed using 3D printing: (a) The appearance of bones right after being 3D printed (b) Assembled result of the printed bones.**

HoloLens 2 Voice was obtained for the functionality of HoloLens 2 Voice. The initial screen of the Metaverse platform provides the ability to create rooms where you can interact with other users or join already created rooms. The 'Create Room' feature is launched by clicking the 'Create Room' button in the 'Sharing Session Panel', then follow these steps: First, check if the Provider exists. Once confirmed, we check the number of objects created through the remote rendering service. Once rendered objects are found, remove all objects and run the Provider's room creation function. The Provider's room creation feature checks the names of all rooms created on the currently connected server. If a room with the same name does not exist, create a new room using the room creation feature.

The 'Join Room' feature follows a similar process. When selecting a room to join, the room name set is compared to the existing room. If a matching room is found, the user joins that room. Additionally, when the model location sharing feature is activated, it is integrated with Azure Anchor Service using the configured ID, key, and domain values. This allows remotely rendered models to exist in the same location, with the model's anchor point relative to the person who created the room.

In metaverse, the functionality of remote rendering is available for mutual use. Upon creating a room and connecting to the Photon server, the current list of connected players is displayed on the right side. In the collaboration panel, the present connection status, room name, and the number of connected players is indicated.

## XR contents

**Menu.** Table 2 presenting a compilation of menu and feature-related items. The "App Menu" incorporates diverse functionalities, including but not limited to tutorial, button selection, menu language change, tool function, partial palette function, description window function, collaboration function, and other menu options. The "Name" entry addresses the display of names based on language options. Fig 2 shows the menu screen of the manipulation tool. In the manipulation tool, users can move or detach bones to display them. Additionally, bones can be differentiated by changing colors, and there is a feature to output the surface of the bones if needed. Fig 3 shows an individual engaging with the XR content firsthand while donning the HoloLens.

## Functions

Table 3 presenting elements related to tool functions, partial palette, description window, and collaboration function. "Tool Functions" encompasses a range of functionalities such as moving the entire model, moving individual parts, cross-section cutting, changing color, and undo

**Table 2. The functionality within the "App Menu" section of XR content.**

| Element | Description |
|---|---|
| App Menu | Application menu including the following items: |
| | Tutorial |
| | Button Selection |
| | Menu Language Change |
| | Tool Function |
| | Partial Palette Function |
| | Description Window Function |
| | Collaboration Function |
| | Other Menu Options |

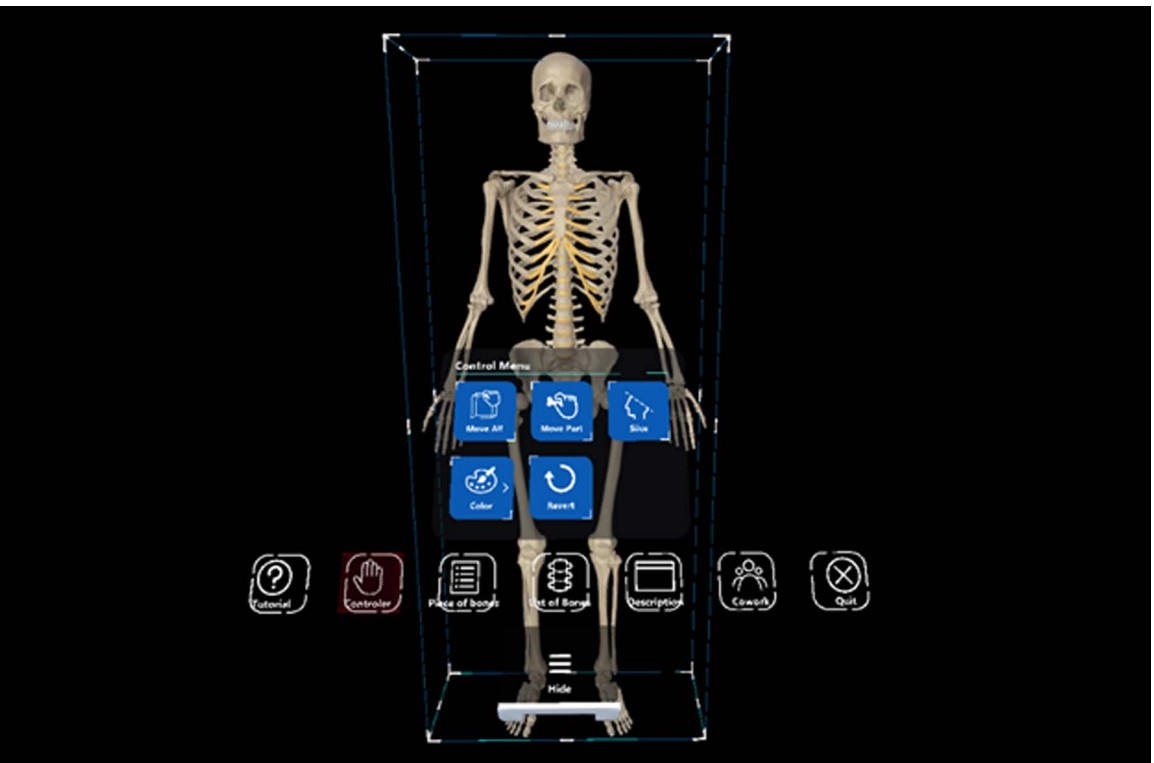

**Fig 2. Menu screen of the manipulation tool. The manipulation menu includes functions to select, move, separate, and add colors to the desired bones.**

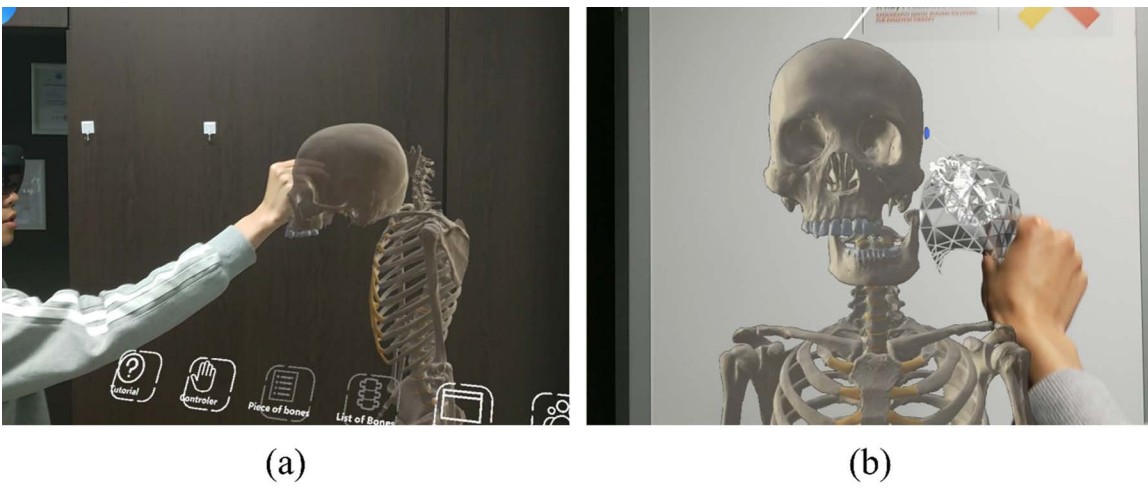

(a)                                                                                  (b)

**Fig 3. Menu screen of the manipulation tool. The manipulation menu includes functions to select, move, separate, and add colors to the desired bones. (a) Side view, (b) Front view.**

function. "Partial Palette" covers various functionalities related to specific body areas. The "Description Window" provides detailed images and descriptions for items within the partial palette. Finally, "Collaboration Function" includes features like creating a room, joining a room, sharing or canceling model location, and ending or exiting the collaboration session.

**Table 3. Various elements incorporated in XR content.**

| Element | Description |
|---|---|
| Tool Functions | Various tool functions including: |
| | Move Entire Model |
| | Move Individual Parts |
| | Cross Section Cutting |
| | Change Color |
| | Undo Function |
| Partial Palette | Various partial palette functions, including the following areas: |
| | Head |
| | Shoulders and Arms |
| | Chest |
| | Pelvis and Legs |
| | Spine |
| | Hands |
| | Feet |
| | Full Body |
| | Human Run |
| | Human Idle |
| Description Window | Window providing detailed images and descriptions for items in the partial palette |
| Collaboration Function | Create Room |
| | Join Room |
| | Share/Cancel Model Location |

## Pilot test feedback

**Group of anonymous college professors.** College professors expressed interest in the applicability of the Metaverse platform across different departments. Majority of their inquiries focused on how the platform could be integrated into school curricula and whether additional content developments would occur to accommodate specific medical departments like ophthalmology, dentistry, cardiology or even veterinary medicine.

**Group of anonymous students.** A participant in an open-ended survey among nursing students who engaged with our program remarked, "Memorizing bone structures from a single diagram has always been a challenge for me. That's why my friends and I always draw diagrams and color them to memorize them for exams. This metaverse program, however, is very fun interacting with each part of the bone. It's like floating bones with quiz cards." Numerous students expressed appreciation for the interactive elements and the capability for real-time collaboration. Another student noted, "When it comes to learning anatomy, I think it's crucial for us to quiz each other and make the learning process more efficient. However, during the COVID-19 restrictions, it was extremely difficult for us to study because none of us were able to study in groups. If we had been provided with this program, I believe it would have been much easier and more enjoyable to have a group of our classmates join a session together to interact with the same cadaver in real-time."

**Group of anonymous corporate guests.** Although the research and pilot testing were not specific intended for buyers, many corporate guests expressed interest in the pricing of available content on the platform and the overall timeline for the development of the upcoming contents. Meanwhile, some corporate guests were particularly fascinated by the originality of the content and wanted to know if major companies or organizations were

already using it. Some reported frequent lags between movements in the scene, which may be due to the quality of internet speed depending on the environment's connection. This highlights the need for optimize the connection between the platform and the internet to ensure a seamless user experience, regardless of internet speed and stability.

**Feedback from the demo session at a medicine conference.** Feedback from the demo session at an medicine conference highlights the ultimate benefits and necessity of the platform in fields that require an accurate understanding of human anatomy, such as Oriental medicine, particularly acupuncture. Users found that viewing the human skeleton in 3D improved comprehension and teaching performances. They expressed fascination with the interactive features that allowed them to zoom in, zoom out, and rotate the skeletal model. All participants at the demo session were optimistic about this research, envisioning potential collaboration between Western and Asian medicine to form a creative, integrated solution in medical field.

AR elements have been evaluated as effective in improving learners' concentration and immersion. This is a huge advantage in educational settings, promoting better engagement and understanding. The platform's potential to replace traditional hands-on methods using live cadaver dissections or physical models was highly praised for its efficiency. Users highlighted benefits such as cost savings, solving hygiene issues and easing constraints in the physical practice environment.

## Discussion

In the development and application of our novel de novo 3D system for anatomical education, it is important to compare its efficacy and advantages against other established 3D and 2D systems used in teaching anatomy. Although our system presents unique features, understanding its relative performance is crucial for assessing its potential impact.

Physical 3D printed models, such as those utilized in cardiology training, offer valuable feedback and interactive experiences that enhance the learning process. Studies have demonstrated that these models can significantly aid in developing a practical understanding of complex anatomical structures [25]. For example, physical models allow learners to engage in hands-on manipulation, which is instrumental in reinforcing spatial relationships and anatomical details.

In the field of otolaryngology, 3D printed models created from cadaveric temporal bones, derived from micro-CT scans, are employed for surgical training. These models highlight the advantages of tactile interaction, providing a more immersive and practical training experience [26]. Such models help trainees appreciate the texture and physicality of anatomical structures, which can be crucial for surgical precision and technique.

Further, research on virtual reality (VR) and augmented reality (AR) systems reveals their potential in enhancing spatial understanding and increasing student engagement. VR and AR technologies offer immersive experiences that can complement traditional learning methods by providing dynamic and interactive visualizations of anatomical structures [27]. These technologies allow learners to explore and manipulate 3D models in a virtual environment, which can be particularly beneficial for complex anatomical education.

When compared to traditional 2D systems, our de novo 3D system offers several distinct advantages. Traditional 2D images, such as those found in textbooks or standard medical illustrations, often fail to convey the spatial relationships and depth of anatomical structures effectively. This limitation can hinder a learner's ability to grasp the three-dimensional nature of anatomy, which is essential for understanding spatial orientation and performing clinical procedures.

In contrast, our 3D system provides a comprehensive, three-dimensional perspective of anatomical structures, allowing users to interact with and view models from multiple angles. This capability enhances spatial comprehension and offers a more intuitive understanding of anatomical relationships. Moreover, unlike static 2D images, the dynamic and interactive

nature of 3D models facilitates a deeper engagement with the material, supporting more effective learning and retention.

The utilization of 3D scanning and modeling techniques results in the creation of high-fidelity interactive 3D DT models with precise anatomical information for deeper learning experiences. The incorporation of technologies such as Photon Network facilitates real-time collaboration among students, fostering deeper engagement, and facilitating a collaborative educational environment. Additionally, the adoption of cloud-based remote rendering technology ensures seamless and high-quality visualization of 3D models across diverse devices, contributes to cost reduction in hardware and enhancing accessibility.

The final product of this novel development provides valuable insight towards the future of anatomy education, highlighting the pivotal role of XR technology and 3D DTs in offering boundless learning opportunities for users. Ethical considerations related to the development and use of 3D digital twins are emphasized, addressing concerns associated with practical anatomy education and underscoring a commitment to upholding ethical standards.

The results present the need for improvement and scalability, with the technology having the potential to expand and encompass more anatomical content, finding application in various medical fields and offering a versatile educational platform. XR technology, including the metaverse, emerges as a transformative force in anatomy education, suggesting potential replacements for traditional cadaver dissection, enhanced accessibility, and efficient learning experiences, particularly through devices like HoloLens 2.

The pilot test feedback gathered in this study indicates widespread interest in potential applications across various fields, from education to business within medicine. Key improvements identified include addressing latent challenges related to internet connectivity and lagging, developing tailored content for specific departments, and enhancing platform integration for further business utilization. Additionally, 100% of the pilot test feedback from participants were positive regarding the platform and content's interactive and immersive features, highlighting its potential to open new doors to traditional learning and training methods in medicine. Moving forward, these valuable insights collected will serve as a steppingstone for further development and optimization of the platform to enhance the complementary research based on the 3D DT of the KSMC. A short video (S1 Video.) showcases few moments from pilot tests and provides a walkthrough of the developed Metaverse platform and the 3D DT of the KSMC, called 'Skeletomy.' This video, included as supplementary material, provides an immersive experience from various perspectives. Two users are featured in the video: the main user (female) is shown from a 1st person perspective, and the secondary user (male) is shown in 3rd person perspective. Together, they demonstrate real-time collaboration features as they interact with the same 3D digital twin of the Korean Standard Male Cadaver. Through this video, viewers also get a chance to immerse themselves in an indirect experience of learning with the KSMC and to envision a new virtual era for applications in anatomy education and telemedicine.

Amidst the high performance of this novel development, several areas seemed to need improvement for complementary research. As shown in Fig 4 (a), the length and thickness of the sutures at the junctions between the lumbar vertebrae and the torso, as well as between the torso and the head, were found to be insufficient and required reinforcement. Although there were no issues with support, there was a tendency for swaying due to the weight distribution between the torso and the head bones. Next, as depicted in Fig 4 (b), there was a need to change the material of the rib cartilage to reduce the risk of damage. To achieve a connection between different materials, the model will be reworked using a suturing method. Lastly, as illustrated in Fig 4 (c), there was a need to address the unstable structure where the wing bones were only connected to the chest bones. To improve this aspect in the future, additional connection points between the rib bones and the wing bones will be incorporated.

The model's junctions include the skull-neck, neck-torso, shoulder-upper arm, upper arm-lower arm, lower arm-wrist, torso-lumbar vertebrae, lumbar vertebrae-thigh bone, thigh bone-shin bone, shin bone-ankle bone, etc., as demonstrated in Fig 5. The detailed enhancement plan for the 3D skeletal cadaver is outlined as follows.

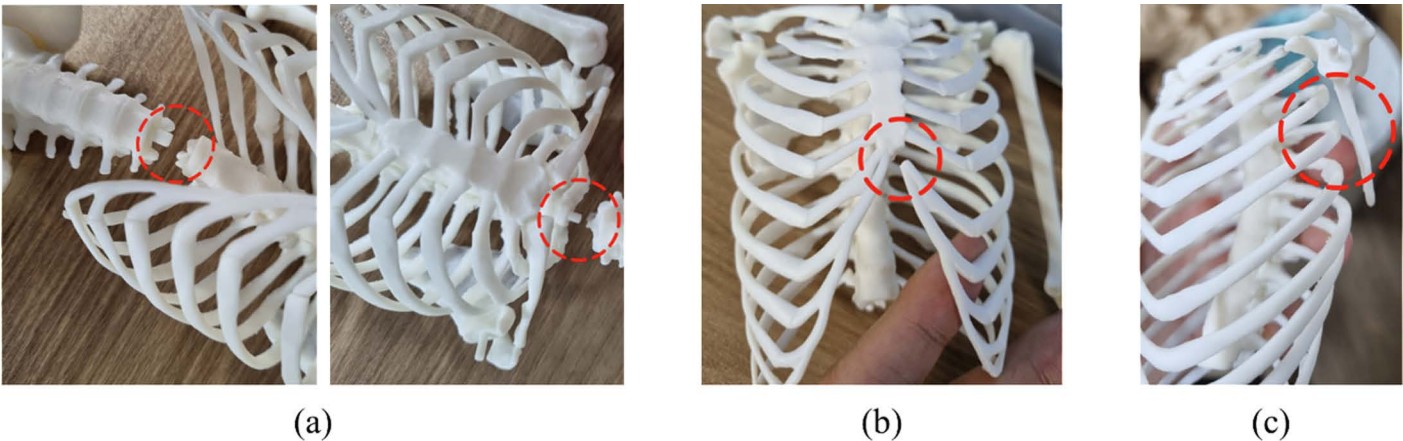

(a)                (b)                (c)

**Fig 4. The content screen corresponding to the "Partial Palette" in the "App Menu. " "Partial Palette" allows the selection of specific bone regions and provides options to observe the movements of the human body.**

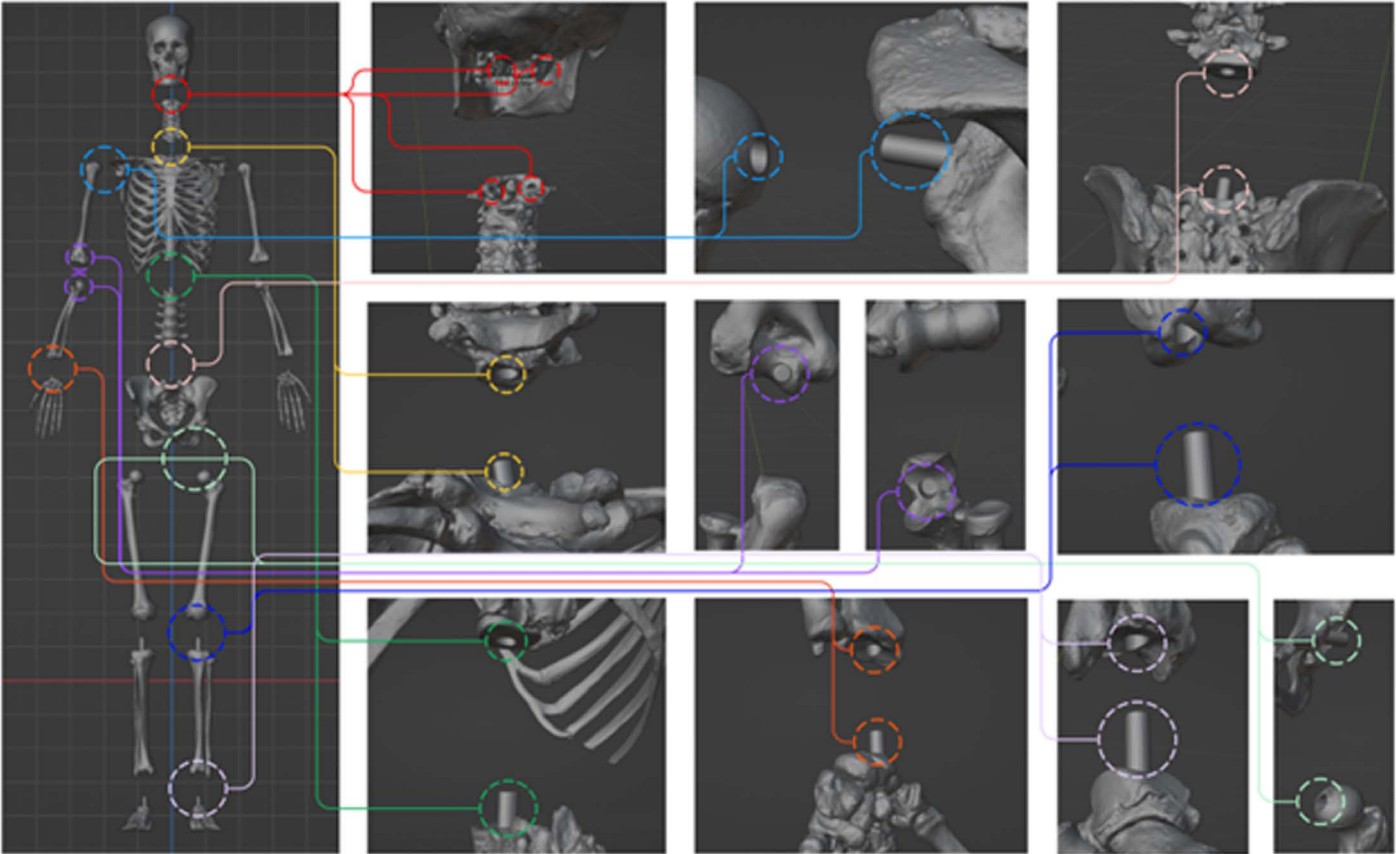

**Fig 5. Modeling of articulations for the creation of a 3D skeletal model.**

The exploration of XR anatomy education revealed specific technical challenges, which were manifested by the process bone scanning and modeling processes. In addition, while such novel XR educational tool demonstrates a significant advancement in medical education, challenges such as battery life, high costs, and internet connectivity persist. Nonetheless, these developments mark a substantial leap towards a new era in collaborative and interactive learning, ensuring improved performance, user experience, and unlimited digital access. Not only do they facilitate the creation of interactive 3D DTs from MRI or CT images, expanding beyond gross anatomy to specialized curricula, but also open opportunities in integration with AI and machine learning to enhance educational content based on data-driven insights and user experience. The positive outcomes underscore the promising role of XR technology in healthcare and advocate for further innovative applications.

## Conclusion

Technological innovation in the field of anatomy education is poised to play a crucial role in shaping the future generation of medical and healthcare professionals. The proposed content in this study holds significance in overcoming the limitations of traditional anatomy education and spearheading innovation in the learning experiences of future practitioners. Sustained efforts in the development and enhancement of such technologies are imperative, as they are poised to furnish learners with essential tools for success in the evolving landscape of healthcare in the future.

## Supporting information

**S1 Video. A short video demonstrating a walkthrough of the developed Metaverse platform and the 3D DT of the KSMC of this research, 'Skeletomy.**' showcasing an immersive experience from various perspctives. Together, they demonstrate real-time collaboration features as they interact with the same 3D digital twin (DT) of the Korean Standard Male Cadaver (KSMC). Through this video, viewers also get a chance to immerse themselves in an indirect experience of learning with the KSMC and to envision a new virtual era for applications in anatomy education and telemedicine.
(MP4)

## Author contributions

**Conceptualization:** So Hyun Ahn, Seung Ho Han.

**Formal analysis:** Seo Yi Choi, Dong Hyeok Choi, Sung Ho Cho.

**Resources:** Sung Ho Cho, So Hyun Ahn, Seung Ho Han.

**Supervision:** So Hyun Ahn, Seung Ho Han.

**Writing – original draft:** Seo Yi Choi, Dong Hyeok Choi.

**Writing – review & editing:** Seo Yi Choi, Dong Hyeok Choi, So Hyun Ahn, Seung Ho Han.

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
