## [Decision Letter · Decision Letter 0]

2 Jul 2024

PONE-D-24-22363Development and Utilization of 3D Anatomy Education Content using Metaverse and XR for Remote Telemedicine EducationPLOS ONE

Dear Dr.  Ahn, Thank you for submitting your manuscript to PLOS ONE. After careful consideration, we feel that it has merit but does not fully meet PLOS ONE’s publication criteria as it currently stands. Therefore, we invite you to submit a revised version of the manuscript that addresses the points raised during the review process.

We look forward to receiving your revised manuscript.

Kind regards,

Priti Chaudhary, M.S.

Academic Editor

PLOS ONE

Journal Requirements:

Additional Editor Comments:

Authors are required to reply all the queries, raised by both the reviewers.

Please add the document related to Ethical waiver from Institutional Ethics Committee/ Research Advisory Committee.

Reviewers' comments:

Reviewer's Responses to Questions

Comments to the Author

1. Is the manuscript technically sound, and do the data support the conclusions?

Reviewer #1: Yes

Reviewer #2: Yes

2. Has the statistical analysis been performed appropriately and rigorously? 

Reviewer #1: N/A

Reviewer #2: N/A

3. Have the authors made all data underlying the findings in their manuscript fully available?

Reviewer #1: Yes

Reviewer #2: Yes

4. Is the manuscript presented in an intelligible fashion and written in standard English?

Reviewer #1: Yes

Reviewer #2: Yes

5. Review Comments to the Author

Reviewer #1: introduction: informative but the main aim of the research should be briefly addressed at its end

methedology: presented with detailed structure

results: great effort is recognized adressing short video for the actual usage of the system would have been a marked

result.

- addressing any reviews of the users either students or professors would have been a plus OR it may be a

complementary research work???!!!

discussion: though it is clear that it is a denovo system but i would have prefered to compare it to other 3d systems that

may be used in teaching anatomy, or address its advantages to well known 2d systems and so on

Reviewer #2: Dear author

I really appreciate the effort done in replicating the twin model and the sophistication of methodology listed, but I have a few questions that I couldn't get past regarding the delivered manuscript

1- Where is the "Korean Standard Full-body Cadaver in metaverse", all what was described was merely bone scans, no flesh, no organs, no full body

2- You mentioned that this project was assessed by a cohort of medical professionals and students from Ewha Woman’s University and 40 other nursing schools in Korea, where is this assessment

3- The name of the study implies the use of metaverse/XR technology to replace traditional teaching with its limitations, why the need for a 3D printing model for bones of the body from the start

4- What's the importance of 3D printing for the bone scans, despite the fact that plastic models deliver a reasonable accuracy for this particular point that alleviates the need for a 3D print of it.

Apart of these comments, the rational is acceptable, the methodology is detailed and properly enlisted, but no field studies were added to assess the validity of the proposed tool in anatomical education

6. PLOS authors have the option to publish the peer review history of their article (what does this mean? ). If published, this will include your full peer review and any attached files.

**Do you want your identity to be public for this peer review?** For information about this choice, including consent withdrawal, please see our Privacy Policy .

Reviewer #1: No

Reviewer #2: No

---

## [Author Response · Author response to Decision Letter 1]

2 Jan 2025

Reviewer #1:

1. introduction: informative but the main aim of the research should be briefly addressed at its end

Thank you for your comment.

The main purpose of the study has been added at the end of the introduction.

L116:

In essence, this research aims to pioneer XR anatomy education developing a 3D DT of a KSMC in the metaverse, introducing transformative and collaborative approaches to current challenges and future possibilities in medical education and telemedicine.

2. methodology: presented with detailed structure

Thank you for your comment regarding the detailed structure of the methodology section. We have made further enhancements to the clarity and comprehensiveness of the "Materials and Methods" section to ensure it accurately conveys the process and techniques used in developing the 3D Digital Twin Cadaver and the accompanying XR content for the metaverse platform.

Before revision of the version methodology structure is as follows.

MATERIALS AND METHODS

From Cadaver to a 3D Digital Twin Cadaver

Collecting and Scanning

Refining and Retopologizing the Mesh for Accuracy

Optimizing Polygons

Architecture for the metaverse platform to Facilitate a 3D digital twin

HoloLens 2

Developing a Metaverse Platform and Adding Interactivity

Development of XR contents

The revised methodology structure is as follows:

MATERIALS AND METHODS

From Cadaver to a 3D Digital Twin Cadaver

Collecting and Scanning

Refining the Mesh

Retopologizing the mesh

Optimizing Polygons

The architecture to facilitate the 3D DT in metaverse

HoloLens 2

Developing a Metaverse Platform

Developing real-time interaction features between users

Enhancing XR content with remote rendering

Pilot test feedback

3. results: great effort is recognized addressing short video for the actual usage of the system would have been a marked

We sincerely appreciate your suggestion and believe that a video can be an effective supplement to our Manuscript.

L414:

A short video showcases few moments from pilot tests and provides a walkthrough of the developed Metaverse platform and the 3D DT of the KSMC, called ‘Skeletomy.’ This video, included as supplementary material, provides an immersive experience from various perspectives. Two users are featured in the video: the main user (female) is shown from a 1st person perspective, and the secondary user (male) is shown in 3rd person perspective. Together, they demonstrate real-time collaboration features as they interact with the same 3D digital twin (DT) of the Korean Standard Male Cadaver (KSMC). Through this video, viewers also get a chance to immerse themselves in an indirect experience of learning with the KSMC and to envision a new virtual era for applications in anatomy education and telemedicine.

4. Result.- addressing any reviews of the users either students or professors would have been a plus OR it may be a complementary research work???!!!

Per your suggestion, we added some quotes and comments from anonymous reviews and feedback that we gathered from students, professors, and professionals in the medical field. We agree that adding these would enhance the solidity of this article.

Consequently, we have included several anonymous reviews and feedback—not only highlighting honest pros and cons but also offering perspectives on potential improvements for complementary research.

L246:

Pilot test feedback

The Metaverse platform and the 3D DT content developed in our study underwent a thorough process of gathering pilot test feedback from anonymous participants in the medical field, including company employees, students and professors from medical and nursing schools. This feedback was collected through an open-ended survey covering general inquiries and interests, evaluations of quality and potential adoptability, and suggestions for further improvements. No participants were required to answer all categories, resulting in some respondents addressing one, two, or all of the topics.

L317:

Pilot test feedback

Group of anonymous college professors

College professors expressed interest in the applicability of the Metaverse platform across different departments. Majority of their inquiries focused on how the platform could be integrated into school curricula and whether additional content developments would occur to accommodate specific medical departments like ophthalmology, dentistry, cardiology or even veterinary medicine.

Group of anonymous students

A participant in an open-ended survey among nursing students who engaged with our program remarked, “Memorizing bone structures from a single diagram has always been a challenge for me. That’s why my friends and I always draw diagrams and color them to memorize them for exams. This metaverse program, however, is very fun interacting with each part of the bone. It’s like floating bones with quiz cards.” Numerous students expressed appreciation for the interactive elements and the capability for real-time collaboration. Another student noted, “When it comes to learning anatomy, I think it’s crucial for us to quiz each other and make the learning process more efficient. However, during the COVID-19 restrictions, it was extremely difficult for us to study because none of us were able to study in groups. If we had been provided with this program, I believe it would have been much easier and more enjoyable to have a group of our classmates join a session together to interact with the same cadaver in real-time.”

Group of anonymous corporate guests

Although the research and pilot testing were not specific intended for buyers, many corporate guests expressed interest in the pricing of available content on the platform and the overall timeline for the development of the upcoming contents. Meanwhile, some corporate guests were particularly fascinated by the originality of the content and wanted to know if major companies or organizations were already using it. Some reported frequent lags between movements in the scene, which may be due to the quality of internet speed depending on the environment’s connection. This highlights the need for optimize the connection between the platform and the internet to ensure a seamless user experience, regardless of internet speed and stability.

Feedback from the demo session at a medicine conference

Feedback from the demo session at an medicine conference highlights the ultimate benefits and necessity of the platform in fields that require an accurate understanding of human anatomy, such as Oriental medicine, particularly acupuncture. Users found that viewing the human skeleton in 3D improved comprehension and teaching performances. They expressed fascination with the interactive features that allowed them to zoom in, zoom out, and rotate the skeletal model. All participants at the demo session were optimistic about this research, envisioning potential collaboration between Western and Asian medicine to form a creative, integrated solution in medical field.

AR elements have been evaluated as effective in improving learners' concentration and immersion. This is a huge advantage in educational settings, promoting better engagement and understanding.

The platform's potential to replace traditional hands-on methods using live cadaver dissections or physical models was highly praised for its efficiency. Users highlighted benefits such as cost savings, solving hygiene issues and easing constraints in the physical practice environment.

5. discussion: though it is clear that it is a denovo system but i would have prefered to compare it to other 3d systems that may be used in teaching anatomy, or address its advantages to well known 2d systems and so on

We are grateful for the insightful feedback provided by the reviewer. Your comment highlighted the importance of contextualizing our de novo 3D system within the broader landscape of anatomical teaching tools. In response, we have added a comparative analysis to the discussion section of our manuscript.

Specifically, we have included a comparison of our 3D system with other 3D models, such as physical 3D printed models used in cardiology and otolaryngology training, and with VR and AR technologies. This addition emphasizes the tactile advantages and immersive experiences offered by these systems (25, 26, 27).

Additionally, we addressed the advantages of our 3D system compared to traditional 2D systems, highlighting its ability to provide a more comprehensive and interactive understanding of anatomical structures, which is often lacking in 2D representations.

L359:

In the development and application of our novel de novo 3D system for anatomical education, it is important to compare its efficacy and advantages against other established 3D and 2D systems used in teaching anatomy. Although our system presents unique features, understanding its relative performance is crucial for assessing its potential impact.

Physical 3D printed models, such as those utilized in cardiology training, offer valuable feedback and interactive experiences that enhance the learning process. Studies have demonstrated that these models can significantly aid in developing a practical understanding of complex anatomical structures (25). For example, physical models allow learners to engage in hands-on manipulation, which is instrumental in reinforcing spatial relationships and anatomical details.

In the field of otolaryngology, 3D printed models created from cadaveric temporal bones, derived from micro-CT scans, are employed for surgical training. These models highlight the advantages of tactile interaction, providing a more immersive and practical training experience (26). Such models help trainees appreciate the texture and physicality of anatomical structures, which can be crucial for surgical precision and technique.

Further, research on virtual reality (VR) and augmented reality (AR) systems reveals their potential in enhancing spatial understanding and increasing student engagement. VR and AR technologies offer immersive experiences that can complement traditional learning methods by providing dynamic and interactive visualizations of anatomical structures (27). These technologies allow learners to explore and manipulate 3D models in a virtual environment, which can be particularly beneficial for complex anatomical education.

When compared to traditional 2D systems, our de novo 3D system offers several distinct advantages. Traditional 2D images, such as those found in textbooks or standard medical illustrations, often fail to convey the spatial relationships and depth of anatomical structures effectively. This limitation can hinder a learner's ability to grasp the three-dimensional nature of anatomy, which is essential for understanding spatial orientation and performing clinical procedures.

In contrast, our 3D system provides a comprehensive, three-dimensional perspective of anatomical structures, allowing users to interact with and view models from multiple angles. This capability enhances spatial comprehension and offers a more intuitive understanding of anatomical relationships. Moreover, unlike static 2D images, the dynamic and interactive nature of 3D models facilitates a deeper engagement with the material, supporting more effective learning and retention.

25. Chetan D, Valverde I, Yoo S-J. 3D Printed Models in Cardiology Training. COLLEGE OF CARDIOLOGY FOUNDATION 2024;3.

26. Lähde S, Hirsi Y, Salmi M, Mäkitie A, Sinkkonen ST. Integration of 3D-printed middle ear models and middle ear prostheses in otosurgical training. BMC Medical Education. 2024.

27. Zulfiqar F, Raza R, Khan MO, Arif M, Alvi A, Alam T. Augmented Reality and its Applications in Education: A Systematic Survey. IEEE Access. 2023;11:143250-71.

Reviewer #2:

I really appreciate the effort done in replicating the twin model and the sophistication of

methodology listed, but I have a few questions that I couldn't get past regarding the delivered

manuscript

6. Where is the "Korean Standard Full-body Cadaver in metaverse", all what was described was merely bone scans, no flesh, no organs, no full body

Thank you for your valuable comments. As noted, our current research is focused on the initial phase of creating a digital twin from the bone scan of a Korean male cadaver. At this stage, the 3D dataset includes only the bone structures, and does not yet incorporate flesh, organs, or other body parts. Future studies are planned to include these additional anatomical features.

To clarify, the primary goal of this research is to explore complementary approaches for enhancing anatomy education and expanding telemedicine applications. While the initial phase concentrates on the bone data, subsequent research will aim to develop a more comprehensive full-body model.

L100:

Key aspects include the use HoloLens 2 Augmented Reality Head-Mounted Display (AR HMD) technology and real-time cloud rendering for an immersive and interactive educational experience. Initial 3D datasets, comprising extensive anatomical information for each bone, were obtained through the 3D scanning of a full-body cadaver of a Korean male origin. This initial study focuses on the digital twin created from the bone scan of the Korean cadaver, with complementary studies planned to include flesh, organs, and other body parts.

7. You mentioned that this project was assessed by a cohort of medical professionals and students from Ewha Woman’s University and 40 other nursing schools in Korea, where is this assessment.

Thank you for pointing that out.

Since our experiment gathered only open-ended surveys and anonymous feedback from students, professors, and professionals in the medical field, we have revised and specified the part that originally stated “cohort” to “open-ended survey” and “feedback.” Anonymous feedback and quotes have been added to strengthen the validity and clarity of our findings. Accordingly, to ensure anonymity the previously mentioned introduction, “from Ewha Woman’s University and 40 other nursing schools,” has been revised to “from medical professionals and students from nursing schools.”

L136:

Once the 3D DT of the KSMC was successfully imported into the developed Metaverse platform, the pilot test feedback was gathered from anonymous participants in the medical field, including company employees, students and professors from medical and nursing schools. They included open-ended survey from interests, evaluations and adoptability, and suggestions for improvements, which will be found throughout the article. No participants were required to answer all categories but required to answer at least one.

L317:

Pilot test feedback

Group of anonymous college professors

College professors expressed interest in the applicability of the Metaverse platform across different departments. Majority of their inquiries focused on how the platform could be integrated into school curricula and whether additional content developments would occur to accommodate specific medical departments like ophthalmology, dentistry, cardiology or even veterinary medicine.

Group of anonymous students

A participant in an open-ended survey among nursing students who engaged with our program remarked, “Memorizing bone structures from a single diagram has always been a challenge for me. That’s why my friends and I always draw diagrams and color them to memorize them for exams. This metaverse program, however, is very fun interacting with each part of the bone. It’s like floating bones with quiz cards.” Numerous students expressed appreciation for the interactive elements and the capability for real-time collaboration. Another student noted, “When it comes to learning anatomy, I think it’s crucial for us to quiz each other and make the learning process more efficient. However, during the COVID-19 restrictions, it was extremely difficult for us to study because none of us were able to study in groups. If we had been provided with this program, I believe it would have been much easier and more enjoyable to have a group of our classmates join a session together to interact with the same cadaver in real-time.”

8. The name of the study implies the use of metaverse/XR technology to replace traditional teaching with its l

---

## [Editor Report · Decision Letter 1]

17 Jan 2025

Anatomy Education Potential of the First Digital Twin of a Korean Cadaver

PONE-D-24-22363R1

Dear Dr.Sohyun Ahn,

We’re pleased to inform you that your manuscript has been judged scientifically suitable for publication and will be formally accepted for publication once it meets all outstanding technical requirements.

Kind regards,

Priti Chaudhary, M.S.

Academic Editor

PLOS ONE
---

## [Editor Report · Acceptance letter]

PONE-D-24-22363R1

PLOS ONE

Dear Dr. Ahn,

I'm pleased to inform you that your manuscript has been deemed suitable for publication in PLOS ONE. Congratulations! Your manuscript is now being handed over to our production team.

Kind regards,

on behalf of

Dr. Priti Chaudhary

Academic Editor

PLOS ONE